# The Impact of ChatGPT on English for Academic Purposes (EAP) Students' Language Learning Experience: A Self-Determination Theory Perspective

Jinming Du [1,2,*] and Antonie Alm [1]

1   Languages and Cultures Programme, School of Arts, University of Otago, 95 Albany Street, Dunedin 9016, New Zealand; antonie.alm@otago.ac.nz
2   Higher Education Development Centre, University of Otago, 65-75 Union Place West, P.O. Box 56, Dunedin 9016, New Zealand
*   Correspondence: duji6714@student.otago.ac.nz

**Abstract:** This qualitative study explores the perceptions of English language students regarding the use of the generative AI tool, ChatGPT, as a supportive tool for English for Academic Purposes (EAP) students in a New Zealand university context. Using self-determination theory (SDT) as an explanatory framework, this study explores how ChatGPT impacts students' basic psychological needs for autonomy, competence, and relatedness in their language-learning experience. Semi-structured interviews are conducted with 24 postgraduate EAP students and the data are analysed using thematic analysis. The findings suggest that ChatGPT has the potential to support students' needs for autonomy and competence by providing flexibility, personalised feedback and a safe space for practice. However, the impact on relatedness needs is mixed, with some students experiencing a sense of companionship and others expressing concerns about reduced human interaction. While students acknowledge the benefits of ChatGPT, they also emphasise the importance of human-teacher interactivity and empathy. The findings provide theoretical insights and practical recommendations for educators seeking to integrate generative AI tools effectively into language education.

**Keywords:** ChatGPT; generative AI; EAP; self-determination theory

## 1. Introduction

The recent progress in the field of generative artificial intelligence (GenAI) and large language models (LLMs) has led to the development of powerful tools like ChatGPT (Generative Pre-Trained Transformer). The potential of these tools for language education, particularly their ability to provide personalised, contextualised, and interactive responses, has attracted the interest of educators and researchers, including in the field of English for Academic Purposes (EAP).

EAP aims to equip speakers of English as an additional language with the linguistic competencies necessary for success in academic contexts [1]. As the demand for EAP courses continues to grow in higher education institutions worldwide [2], educators are exploring innovative ways to support students' language-learning needs. The integration of GenAI tools like ChatGPT into EAP curricula has the potential to offer learners enhanced opportunities for practice and personalised support.

However, there is a lack of research examining the motivational aspects of students' engagement with this technology, particularly within the context of EAP. Motivation is a key factor in language learning, as it influences learners' willingness to engage, persist, and succeed in their language studies [3]. Self-determination theory, a motivational framework developed by Ryan and Deci [4], posits that satisfying three basic psychological needs—autonomy, competence, and relatedness—is essential for fostering intrinsic motivation, well-being, and optimal functioning.

The present study aims to address this research gap by exploring the impact of Chat-GPT on EAP students' language-learning experience through the lens of SDT. By investigating how the use of ChatGPT in EAP learning interacts with students' basic psychological needs, this research seeks to provide empirical evidence to inform the effective integration of GenAI tools into EAP pedagogy and curriculum design.

The findings of this study provide a motivational perspective on students' experiences with ChatGPT. By understanding how this technology interacts with learners' basic psychological needs, educators can make informed decisions about how to use the technology effectively, while addressing potential challenges. As GenAI continues to develop, ongoing research into its impact on motivation and engagement in language learning will ensure that the integration of these tools supports the fundamental human dimensions of language learning.

## 2. Literature Review

### 2.1. ChatGPT in Language Education

The introduction of ChatGPT has triggered significant interest in the field of language education due to its potential to facilitate learning and offer individualised experiences for students [5,6]. Several studies have investigated the use and impact of ChatGPT in different language-teaching contexts. Kaplan-Rakowski et al. [7] conducted a comprehensive survey of 147 teachers and found that, regardless of their teaching style, teachers were generally positive about ChatGPT. They believed that the tool could contribute to their professional development and be beneficial for students. Similarly, Guo and Wang [8] investigated the potential of ChatGPT to support English as a foreign language (EFL) teachers' feedback on students' argumentative writing. They found that ChatGPT could generate a significant amount of feedback and distribute attention to content, organisation and linguistic aspects, highlighting its potential to support teachers in providing comprehensive feedback.

However, when ChatGPT was first introduced, the response from academics was mixed. Alm and Ohashi [9] surveyed 367 language teachers worldwide and examined their responses within the first ten weeks of ChatGPT's release. The results indicated that most educators had a moderate awareness of ChatGPT and there was a general reluctance to use it for writing feedback or conducting automated assessments. Despite this initial reluctance, researchers have continued to explore the benefits and potential risks of ChatGPT in educational settings [8,10–12].

While much of the initial research on ChatGPT in language education has focused on the perspectives of teachers, some studies have begun to explore the experiences and perceptions of learners. Xiao and Zhi [13] investigated the pedagogical potential of ChatGPT for improving the English language performance of English language students at an international English-medium institution in China. A small qualitative study was conducted to assess the students' views and experiences of using ChatGPT. It led to improvements in the critical evaluation of the quality of students' ideas and their language learning outcomes. Participants valued ChatGPT as an interactive personal tutor and learning tool that provided logical feedback and correction at any time in relation to International English Language Testing System (IELTS) writing learning. This qualitative research revealed that out of five students, three described ChatGPT as a peer tutor, providing support that was perceived to be more accessible compared to the support provided by teachers.

Young and Shishido [14] investigated the suitability of ChatGPT for producing dialogue materials for EFL students. They evaluated the effectiveness of ChatGPT for learning English dialogues by studying a sample of 450 conversations. The results indicate that ChatGPT is highly suitable for students at Common European Framework of Reference for Languages (CEFR) level A2. Participants were able to understand most of the words used during the learning process because they did not encounter colloquial expressions that might confuse English learners. Although this study shows that students can understand most English words, its limitation lies in the fact that only English learners at an elementary language level were assessed.

Tseng and Lin [15] conducted a qualitative study with 15 non-English majors recruited from a private university in Taiwan during the university's "English Composition III" course. The results indicate that ChatGPT significantly increases the efficiency of English writing by providing instant feedback and generating creative content ideas, thereby speeding up the writing process. The study also shows that ChatGPT ensures the coherence of students' writing and guides them to organise their thoughts more logically. It emphasises that when students interact with ChatGPT, they enter into a dynamic partnership.

As research on learners' experiences of ChatGPT continues to emerge, it is important to consider how this technology interacts with students' motivation in language learning. Kasneci et al. [6] highlighted the potential of ChatGPT to accurately assess students' learning barriers and progress, allowing for targeted interventions. Raj et al. [12] highlighted the potential of ChatGPT to automate and improve the grading system, which could streamline the evaluation process for teachers. Alm and Watanabe [10] explored the impact of ChatGPT on language pedagogy through the lens of Paul Freire's critical pedagogy. They highlighted ChatGPT's ability to provide personalised, interactive and situated learning opportunities, while acknowledging the risk of perpetuating cultural biases and facilitating passive learning. However, they also noted that ChatGPT can be guided with specific prompts to encourage dialogue and experiential learning. As research on ChatGPT in language education continues to develop, it is clear that the tool has the potential to transform teaching and learning practices, despite the initial mixed reactions from academics.

### 2.2. English for Academic Purposes (EAP)

English for Academic Purposes (EAP) is a form of English language education that aims to assist speakers of English as an additional language in developing the linguistic competencies essential for scholarly pursuits [1]. The focus on learners in the EAP context has expanded in recent years [16,17], with EAP courses attempting to address the various demands students face in academia [2,17]. EAP programmes focus on improving proficiency in reading, writing, listening, and speaking within the context of academic discourse. ChatGPT has been integrated into EAP classes to facilitate language learning by offering learners interactive activities to enhance their grammar, vocabulary, and sentence structure [18]. Additionally, English learners can use ChatGPT to receive assistance with writing tasks, such as generating ideas, improving sentence clarity, and checking for grammatical errors [19]. Lingard [20] highlighted that ChatGPT can serve as a writing aid or a tool for brainstorming, further supporting EAP students in their academic writing.

### 2.3. Self-Determination Theory (SDT)

Self-determination theory (SDT) is a macro-theory of human motivation, development, and well-being that has been widely applied in various fields, including language education [4]. SDT comprises several mini-theories, of which Basic Psychological Needs Theory (BPNT) and Organismic Integration Theory (OIT) are particularly relevant to understanding motivation in educational contexts.

Basic Psychological Needs Theory (BPNT) posits that individuals have three innate psychological needs: autonomy, competence, and relatedness. When these needs are satisfied, individuals are more likely to experience intrinsic motivation, characterised by engaging in activities for their inherent enjoyment and satisfaction [4,21].

Autonomy refers to the need to experience volition and self-endorsement of one's actions. In language learning, this might manifest as learners having choices in their learning activities or perceiving their learning goals as congruent with their personal values.

Competence pertains to the need to feel effective in one's interactions with the environment and to have opportunities to express and develop one's capacities. For language learners, this could involve experiencing self-efficacy in using the target language or successfully completing challenging linguistic tasks.

Relatedness concerns the need to feel connected to others, to experience a sense of belonging, and to engage in reciprocal care [4]. In language-learning contexts, this might be realised through positive interactions with educators and peers or feeling integrated within a broader language learning community. The concept of relatedness can extend to interactions with technology. For instance, Jeon [22] examined relatedness in the context of learner interactions with a self-directed interactive app, focusing on aspects such as receiving support from the app, feeling attached to it, and relying on the information it provides.

Organismic Integration Theory (OIT) focuses on the process of internalisation and the spectrum of extrinsic motivation. It proposes that extrinsic motivation can vary in its degree of autonomy, ranging from external regulation (least autonomous) to integrated regulation (most autonomous). External regulation refers to behaviours motivated purely by external contingencies. Introjected regulation involves partial internalisation, where external regulations are taken in but not fully accepted as one's own. Identified regulation occurs when the individual recognises and accepts the underlying value of a behaviour. Finally, integrated regulation represents the most autonomous form of extrinsic motivation, where identified regulations are fully assimilated into the self [21].

SDT has been successfully applied to language-learning contexts and has provided insights into the factors that influence learner motivation. Noels et al. [23] found that autonomy-supportive teaching practices were associated with increased intrinsic motivation and self-determined forms of extrinsic motivation in language learners. Alamer [24] found that the satisfaction of basic psychological needs was positively associated with autonomous motivation and language achievement in EFL learners. Fryer and Oga-Baldwin [25] applied SDT to investigate the motivation of Japanese junior high school students learning English. Their findings highlighted the importance of intrinsic motivation in language learning and demonstrated that a sense of competence and the satisfaction of psychological needs were significant predictors of engagement and achievement.

In the domain of computer-assisted language learning (CALL), SDT has been used to explore the motivational impact of various tools and platforms. Alm [26] proposed that Internet-based language-learning environments can be motivating due to their potential to support basic psychological needs. The study outlined how Web 2.0 applications can support learners' needs for relatedness, competence, and autonomy through facilitating interaction, collaborative writing activities, and personalised learning experiences. More recently, Jeon [22] investigated primary school students' usage patterns of a self-directed interactive app for informal EFL learning and how the app supported their basic psychological needs. The study found that students who continuously used the app reported greater satisfaction with regard to their competence and relatedness needs than those who discontinued using the app. Zeng and Fisher [27] used SDT as a framework to explain how the use of a gamified language learning app (Duolingo) impacted learners' motivation. They found that the app's ability to satisfy learners' needs for autonomy and competence played a crucial role in fostering intrinsic motivation for language learning. They also proposed the concept of "motivational transfer," suggesting that motivation generated by using a specific language-learning app can transfer to more general motivation for language learning.

Recent research has begun to explore how AI technologies, such as chatbots, interact with learners' basic psychological needs. Chiu et al. [28] found that the impact of AI-based chatbots on learners' need satisfaction and intrinsic motivation was moderated by both teacher support and student expertise. This suggests that the effectiveness of AI technologies in supporting basic psychological needs may vary depending on individual learner characteristics and the learning context.

In the context of this study, autonomy refers to students' sense of volition and choice in their use of ChatGPT for language learning, both within and outside the classroom. Competence includes students' perceived effectiveness and mastery in utilising ChatGPT to

support their language-learning objectives. Relatedness refers to the need to feel connected to others, to have a sense of belonging, and to care for and be cared for by others [4].

Our study aims to examine relatedness in a broader context than some previous research. Jeon [22] operationalised relatedness specifically in terms of the learner's relationship with a self-directed interactive app, focusing on feelings of support, reliability, and attachment to the app itself. In contrast, our study considers both students' feelings of connection in their interactions with ChatGPT and their sense of belonging within the wider language-learning community, including relationships with peers and educators. This broader conceptualisation is consistent with Ryan and Deci's [4] definition of relatedness in SDT, which includes connections to both technological tools and human interlocutors in the language-learning process. This approach allows us to explore how ChatGPT might influence learners' motivation through the satisfaction of basic psychological needs and the internalisation of extrinsic motivation, providing a robust framework for examining the complex relationship between AI-assisted language-learning tools, learner motivation, and the satisfaction of psychological needs in EAP contexts.

### 2.4. Research Gap and Present Study

While previous studies have explored the applications, benefits, and challenges of integrating ChatGPT into language education [6–8,10], few have examined the motivational dimensions of students adopting AI tools for language learning. The motivational aspects of using ChatGPT for language learning remain largely unexplored, particularly within the context of EAP learning in New Zealand. This study aims to address this gap by investigating the motivational dynamics of EAP learning among international postgraduate students at a university in New Zealand through the lens of SDT. Using a motivational theoretical framework enables more effective facilitation of language learners in acquiring language skills, thereby augmenting learning efficiency and engagement [29].

The present study focuses on the following research question: How do autonomy, competence, and relatedness manifest in the use of ChatGPT within the context of EAP learning?

By addressing this question, this study aims to provide empirical insights into the motivations behind the perception and use of ChatGPT by EAP learners, enabling educators to effectively integrate this technology into their pedagogical practices and create learning environments that meet the motivational needs of language learners.

## 3. Methodology

To achieve this goal, we interviewed 24 international students enrolled in an EAP programme at a university in New Zealand. Since the release of ChatGPT by OpenAI in November 2022 (and ChatGPT4 in April 2023), it has become a popular tool among international students attending EAP classes in this region. Consequently, we intentionally selected this group of international students who use this tool for language learning to participate in our study.

### 3.1. Participants

The study participants comprised 24 postgraduate students enrolled in an EAP class from diverse academic disciplines. Please see Table 1 for demographic information about the participants. Within this cohort, the composition included 15 male and 9 female participants. All of the students had achieved a minimum IELTS score of 5, meeting the lowest criterion for enrolment in EAP courses in New Zealand. This indicates that these students possess a fundamental understanding of English concepts and are capable of engaging in basic communication in English. Their ages ranged from 22 to 24 years. English serves as the medium of instruction at the university. The students participated in academic English courses taught by EAP language instructors. The students provided informed consent to participate in the research. Pseudonyms are used to ensure the anonymity of the participants.

**Table 1.** Demographic information about the participants.

| Demographic Characteristic | | Frequency | Percentage |
|---|---|---|---|
| Gender | Male | 15 | 62.50% |
| | Female | 9 | 37.50% |
| First Language (L1) | Chinese (Hànyǔ) | 11 | 45.83% |
| | Japanese (Nihongo) | 7 | 29.17% |
| | Vietnamese | 3 | 12.50% |
| | Indian (Hindi) | 1 | 4.17% |
| | Korean (Hangugeo) | 2 | 8.33% |
| Postgraduate Programme | Business (marketing, tourism, management, finance, accounting) | 9 | 37.50% |
| | Sciences (food science, computer science, zoology, geology, botany, chemistry) | 6 | 25% |
| | Humanities (laws, performing arts, music, politics, history) | 7 | 29.17% |
| | Health Science (pharmacy, dentistry) | 2 | 8.33% |

We employed purposive sampling to conduct this study. Participants were identified and enlisted based on their expertise, experience, and age in the context of EAP learning. We contacted students enrolled in EAP language classes at this university in New Zealand through email and on social platforms such as Little Red Book (a social media and e-commerce platform in China where users share lifestyle content, study progress, product reviews, and shopping tips) and Instagram. Initially, 33 individuals who met the specified criteria expressed willingness to participate in the project. Ultimately, we conducted interviews with 24 postgraduate students. To safeguard their confidentiality, a pseudonym was allocated to each student and their information was kept confidential. Participants were denoted as P1, P2, P3, ... P24. The recruitment criteria included: (1) participants must be international students from EAP classes, with a background in learning English; (2) participants should use ChatGPT for language learning; and (3) voluntarily agree to participate in the study. The sole exclusion criterion for participants was spending less than three weeks using ChatGPT for EAP English learning. The rationale behind this criterion was to ensure that all of the participants had a relatively extensive duration of ChatGPT usage in their learning activities before being interviewed.

*3.2. Data Collection and Procedures*

Semi-structured interviews were conducted to explore the factors associated with the SDT framework that could impact participants' process of using ChatGPT for learning English. The interview questions were developed based on the three basic psychological needs outlined in SDT: autonomy, competence, and relatedness [21]. A mapping of the interview questions to the SDT constructs is provided in Appendix A.

Semi-structured interviews offer significant advantages over structured or open interviews, as their semi-structured nature allows for detailed and effective content analysis. Open-ended questions were developed to guide reflections beyond assumed responses. The interviews were conducted either on a face-to-face basis or through the online Tencent Meeting system, with the sessions being recorded to ensure a comprehensive record for subsequent documentation and analysis. Each participant was interviewed for approximately 20–30 min, and the interview content was transcribed and coded.

*3.3. Data Analysis*

The Tencent Meeting software (Version 3.25) was used to transcribe the audio recordings verbatim, followed by a meticulous review of all of the transcripts to ensure accuracy. Subsequently, the interview data were subjected to thematic analysis, adhering to the approach outlined in [30]. Qualitative data collected from the semi-structured interviews were analysed inductively [31].

The thematic analysis was conducted using MAXQDA (version 24) software and involved the following steps:

1. Familiarisation: The data collected from the 24 students underwent multiple rounds of reading. Iterative reading of interview data facilitates the development of preliminary interpretations.
2. Generating initial codes: All of the segments related to relatedness, autonomy, and competence were coded. The two researchers developed a preliminary coding scheme based on the SDT framework.
3. Searching for themes: Codes with similar underlying meanings were grouped into common themes. These themes were organised in MAXQDA, identifying three main categories of visibility.
4. Reviewing themes: Following the initial coding, the two coders discussed the coding scheme. They then coded all of the transcripts independently, achieving an intercoder reliability percentage of 98%. The high score can be attributed to the straightforward nature of the themes, which facilitated consensus among the coders. The coders agreed on all of the sub-themes, as shown in Table 2. The only sub-category that was identified by only one coder was "peer pressure," but both coders agreed to retain it in the final analysis.
5. Defining and naming themes: The data snippets were revisited to enhance each thematic category and maintain internal consistency. The codes were then grouped into three main themes: autonomy, competence, and relatedness. Representative responses were extracted.
6. Producing the report: The findings were written up, with representative comments documented and detailed in the Findings section.

**Table 2.** Themes and sub-themes identified in the data.

| BPN | Themes | Frequency |
|---|---|---|
| Autonomy | 1. Flexibility and freedom to learn at own pace | 10 |
| | 2. Ownership and control over learning activities | 11 |
| | 3. Safe space to practice without fear of judgement | 13 |
| | 4. Potential overreliance on ChatGPT (−) | 9 |
| Competence | 1. Personalised feedback and error correction | 41 |
| | 2. Support and context-specific examples | 40 |
| | 3. Increased self-efficacy and confidence in EAP tasks | 19 |
| | 4. Concerns about accuracy and relevance of responses (−) | 11 |
| | 5. Inability to adjust language levels as students progress (−) | 8 |
| Relatedness | 1. Sense of companionship and guidance | 13 |
| | 2. Reduced human interaction and emotional connection (−) | 12 |
| | 3. Lack of social belonging in AI interactions (−) | 15 |
| | 4. Peer pressure (−) | 1 |

In the end, 203 coded texts were categorised into three distinct functional categories, with thirteen sub-themes identified (please see Table 2).

*3.4. Ethical Considerations*

This study was conducted following the ethical guidelines set by the university's research ethics committee. All of the participants provided informed consent before they participated in the study. They were informed about the purpose of the study, the voluntary nature of their participation, and their right to withdraw at any time without consequence. Participants were assured that their personal information would be kept confidential and that their identities would be protected through the use of pseudonyms. The recorded interviews and transcripts were stored securely, with access limited to the research team. Participants were also provided with contact information for the researchers and the university's research ethics committee in case they had any concerns or questions about the study.

**4. Findings**

Our findings are presented through the lens of self-determination theory (SDT), focusing on the three basic psychological needs: autonomy, competence, and relatedness. The data from interviews revealed how ChatGPT influences these needs among postgraduate international students learning English in New Zealand. While these students were enrolled in an English for Academic Purposes (EAP) class, their experiences with ChatGPT extended beyond the classroom, supporting their daily interactions and academic pursuits in an English-speaking environment.

This broad impact is exemplified by P3's statement:

Before ChatGPT, I used chatbots like Mondly for practising English speaking. However, since the introduction of ChatGPT in October [sic] last year, I've become a big fan. I use it daily for learning English, and what I really like is its ability to translate text from images. It's not just for regular English learning; I even use it to order food when I go out, especially when faced with English menus I cannot understand. It's a versatile tool that has penetrated both daily life and academic studies.

This quote reflects all three dimensions of SDT, demonstrating how ChatGPT enables students to take control of their learning (autonomy), empowers them to accomplish tasks and enhance their language skills (competence), and fosters relatedness by allowing students to make connections between their academic and everyday life experiences.

*4.1. Autonomy*

The use of ChatGPT significantly impacted students' sense of autonomy in their English language-learning experiences. Four main themes emerged:

1.  Flexibility and freedom to learn at own pace (*n* = 10). P19 emphasised this aspect: "This flexibility allows learners to plan their study hours without the need to adhere to early mornings, motivating them to maximise their learning in an unconstrained time frame." This flexibility allows students to adjust their learning schedule to their individual preferences and needs. Such self-regulation is central to SDT's conceptualisation of autonomy, as it enables students to engage in learning activities at times when they feel most motivated and capable, potentially leading to more effective and intrinsically motivated learning.

2.  Ownership and control over learning activities (*n* = 11). P13's experience exemplifies this: "Initially, I thought it would mainly help with improving my English speaking, as my spoken English was quite poor. However, after using it, I discovered that it's not only beneficial for speaking but also incredibly helpful for writing. It provides ideas and corrects grammar in my writing." This statement illustrates how ChatGPT enables students to identify and address their specific language-learning needs independently, fostering a sense of ownership over their learning process. This self-directed approach resonates with SDT's emphasis on volitional engagement in learning. By allowing students to explore and expand their learning beyond initial expectations, ChatGPT

supports the internalisation process, potentially moving students towards more self-determined forms of motivation.

3. Safe space to practice without fear of judgement ($n = 13$). P18 shared: "I think it's very effective for improving spoken English. I used to be very nervous speaking English in public, but with this software, I can confidently practice at home without fear of making mistakes." This safe environment is particularly valuable for students who experience language anxiety and may be hesitant to participate in class discussions or presentations. By providing a non-judgmental space for practice, ChatGPT supports students' autonomy in taking risks and experimenting with the language. This aspect corresponds to SDT's notion of autonomy-supportive environments, which allows for self-directed learning.

4. Potential overreliance on ChatGPT ($n = 9$). P22 expressed this concern: "I'm particularly worried about the day when I can't use it anymore because I'm currently a paying member and I pay for my membership every month." This point highlights a potential challenge to long-term autonomy. While ChatGPT supports self-directed learning, there is a risk of developing dependence on the tool, which could potentially undermine autonomous learning skills. From a BPNT perspective, overreliance on ChatGPT might impede the development of autonomous motivation for language learning if students become dependent on the tool rather than internalising the value of language learning itself. Simultaneously, it might interfere with the deep processing and practice necessary for second language development. This highlights the need for integration of AI tools in ways that support, rather than replace, students' inherent capacities for autonomous learning and language acquisition.

These findings suggest that while ChatGPT can significantly enhance students' perceived autonomy in language learning, it also introduces new complexities into how autonomy is experienced and developed. The tool appears to support volitional engagement and self-directed learning, key aspects of autonomy in SDT. However, the potential for overreliance highlights the need for careful integration of such tools in ways that foster, rather than replace, students' inherent capacities for autonomous learning.

*4.2. Competence*

ChatGPT significantly influenced students' sense of competence in English, affecting both their academic language skills and their ability to navigate daily life in an English-speaking country. The data revealed both positive and negative aspects:

1. Personalised feedback and error correction ($n = 41$). P17 explained: "It has helped me a lot in improving my English level and pronunciation. For example, my English used to be very poor and my stress and intonation were unstable. I didn't know whether to stress the beginning or the end of a word. However, ChatGPT can help me correct my stress, and for complex words, it patiently teaches me without criticising me." This non-judgmental feedback appears to create a supportive environment where students feel more capable and confident in their language skills. The personalised nature of the feedback relates to the emphasis of SDT on providing optimal challenges and positive feedback to enhance perceived competence.

2. Support and context-specific examples ($n = 40$). P5 noted: "I can ask the teacher to explain again, and it will not only do so but also provide examples that help me understand the concept better. Its examples immerse me in the context, making the knowledge clearer." The ability of ChatGPT to provide context-specific examples enhances students' understanding and application of language concepts. This is consistent with the proposition of SDT that competence is enhanced when individuals can effectively understand and master their environment.

3. Increased self-efficacy and confidence in language tasks ($n = 19$). P16 shared: "I feel it has helped me improve my English language skills significantly. Firstly, my pronunciation was poor, but it could immediately advise me on how to improve it." This increased confidence in language skills is a key component of competence as defined in SDT.

The immediate feedback and suggestions for improvement provided by ChatGPT seem to foster a sense of effectance, which is crucial for intrinsic motivation.

4.  Concerns about accuracy and relevance of responses (*n* = 11). Some participants expressed concerns about ChatGPT's ability to provide appropriate support for complex language tasks, particularly academic writing. For example, P19 noted: "I feel that, at times, the pronunciation guidance for learning English words is accurate and patient. However, when dealing with my EAP writing, the generated text can be rigid and formulaic." This observation suggests potential limitations in terms of ChatGPT's capacity to enhance competence for higher-level language production tasks. However, it also indicates a possible lack of AI literacy among students, who may not be aware of techniques to prompt different outputs or engage in post-editing. This gap in skills could impact students' perceived competence in effective AI tool use.

5.  Inability to adjust language levels as students progress (*n* = 8). P15 said: "What concerns me the most is that in the future, everyone might output similar content, and I worry that it might default to considering me as someone with lower proficiency." While this concern may be more related to the student's lack of prompting skills and insecurity about their progress, it highlights the importance of developing students' ability to effectively communicate their learning needs to AI tools such as ChatGPT. It also illustrates the emotional aspects of language learning and the need for students to feel reassured about their progress and the adaptability of the tools they use. From an SDT perspective, this points to the importance of providing structure and clear pathways for progression to support competence satisfaction.

These findings suggest that ChatGPT can enhance learners' perceived competence through personalised feedback and increased self-efficacy. However, the data also indicate that students often lack the ability to effectively prompt different outputs or engage in post-editing. Developing these AI literacy skills is essential for learners to fully utilise ChatGPT's competence-supportive features and maintain a sense of competence as they progress to more complex language-learning tasks.

### 4.3. Relatedness

The impact of ChatGPT on students' sense of relatedness was mixed, affecting both their in-class interactions and their broader social experiences as international students:

1.  Sense of companionship and guidance (*n* = 13). P20 explained: "It involves an interactive process, unlike traditional learning software, which makes learning more engaging. It provides timely feedback, just like chatting with a real person, and it truly provides companionship." This suggests that, for some students, ChatGPT can provide a form of parasocial interaction, potentially fulfilling some aspects of the need for relatedness. However, this raises questions about the nature of relatedness in human–AI interactions and whether such interactions can truly satisfy the deep-seated need for human connection that SDT posits as essential for well-being.

2.  Reduced human interaction and emotional connection (*n* = 12). P21 admitted: "To be honest, I feel more isolated because it's my first time abroad and I was hoping to interact more with my classmates outside of class, like studying together." This highlights a significant concern: while ChatGPT may provide some form of interaction, it may also reduce opportunities for genuine human connections. From an SDT perspective, this could potentially thwart the satisfaction of relatedness needs, which are crucial for international students' social and cultural adaptation, as well as their overall well-being and motivation.

3.  Lack of social belonging in AI interactions (*n* = 15). P11 explained: "Interacting with ChatGPT for EAP language learning tends to make me feel more isolated. While it serves as a helpful language aid, the absence of genuine human interaction and the emotional support found in peer collaboration can contribute to a sense of isolation." This highlights the limitations of AI in meeting students' need for relatedness. While ChatGPT can support language learning, it cannot replace the emotional and social

aspects of human interactions that are fundamental for a sense of belonging. This is in line with the emphasis of SDT on the importance of warm, caring relationships for optimal motivation and well-being.

4.  Peer pressure (*n* = 1). P18 articulated: "If students know about this software, there will be greater competition in learning English and future employment. I feel that this software is brilliant and helpful. If everyone learns well, I'm particularly worried that others will be better than me in my studies or after I graduate." Although mentioned by only one participant, this point raises an interesting aspect of relatedness in the context of AI-assisted learning: the potential for new forms of social comparison and pressure. From an SDT perspective, this could be seen as a form of introjected regulation, where behaviour is motivated by internal pressures such as guilt or anxiety. This suggests that the use of AI tools could introduce new dynamics into how students relate to and compare themselves with their peers, potentially impacting their sense of relatedness and autonomous motivation.

These findings highlight the complex and sometimes contradictory effects of ChatGPT on students' sense of relatedness. While it can provide a form of interaction and support, it may also contribute to feelings of isolation and introduce new social pressures. This stresses the need to ensure that AI tools are integrated into language-learning contexts to support, rather than hinder, students' social connections and sense of belonging, which are crucial for intrinsic motivation and well-being according to SDT.

### 4.4. Integration of Basic Psychological Needs

Our findings reveal a complex integration of autonomy, competence, and relatedness in students' engagement with ChatGPT for language learning, both within their academic studies and in their daily lives as international students in an English-speaking country. This integration is consistent with the conceptualisation of these three needs as essential and interdependent aspects of psychological growth and well-being [4].

The interdependence of these needs is evident in our participants' experiences. For instance, P13's comment illustrates how ChatGPT simultaneously supports autonomy and competence: "It provides ideas and corrects grammar in my writing. My grammar used to be awful, but ChatGPT has significantly improved my verb tenses and sentence structure." This demonstrates how the tool facilitates volitional engagement (autonomy) while enhancing skill development (competence).

The role of relatedness in this context is more varied. As Ryan and Deci [4] posit, relatedness supports the internalisation of values and behaviours. Our findings reflect this in the diverse experiences of participants. Some students, like P22, experienced a form of connection with ChatGPT: "I feel more socially connected. I feel like I haven't been abandoned by society." This suggests that, for some students, ChatGPT provided a form of relatedness that supported their autonomous engagement and competence development.

However, this experience was not universal. P21 reported feelings of isolation, highlighting how the lack of human interaction may thwart the relatedness needs of some students, potentially affecting their overall motivation and engagement despite gains in autonomy and competence.

These varied experiences illustrate the significant role of relatedness in supporting autonomous motivation and competence development, as posited by SDT. While ChatGPT appears effective at directly supporting autonomy and competence, its impact on relatedness is less consistent. For some students, the tool seems to provide a form of connection that supports their overall motivation. For others, the lack of human interaction may hinder the internalisation process that relatedness typically facilitates.

Relatedness plays a distinct role in fostering the internalisation of extrinsic motivations and supporting intrinsic motivation [4]. The findings suggest that consideration must be directed towards how relatedness is supported in AI-assisted language learning.

The integration of these needs is further illustrated by P5's experience: "In traditional classrooms, if I didn't understand something, I might hesitate to ask the teacher to repeat

it. But with this AI, I feel comfortable asking for clarification without fear of criticism." This comment demonstrates how the support for autonomy (feeling comfortable to ask questions) and competence (receiving clarification) can work together to enhance the learning experience.

However, the potential overreliance on ChatGPT, as previously expressed by P22, highlights the delicate balance required in satisfying these need. This concern reflects the potential tension between the competence support provided by the tool and the need for autonomous learning skills that can persist beyond the use of the technology.

Our findings highlight the complex interaction of autonomy, competence, and relatedness in the context of AI-assisted language learning. While ChatGPT demonstrates strong potential in supporting autonomy and competence, its impact on relatedness varies among learners. This variation highlights the need for a comprehensive approach to integrating AI tools into language-learning environments, one that considers how to foster all three basic psychological needs to support optimal motivation and learning outcomes.

## 5. Discussion

This study aimed to explore how ChatGPT impacts EAP students' basic psychological needs for autonomy, competence, and relatedness. Our findings offer significant insights into the complex ways in which AI-assisted language-learning tools interact with these needs in the context of EAP.

### 5.1. Contributions to Understanding ChatGPT in EAP

Our study reveals that ChatGPT's impact on EAP students' basic psychological needs is multifaceted and varies among individuals. Unlike previous studies that have primarily focused on the general motivational effects of AI in language learning [6–8], our research provides a detailed understanding of how ChatGPT specifically supports or thwarts each of the three basic needs identified in SDT [4].

A key finding is the strong support ChatGPT provides for autonomy and competence in EAP contexts. Participants consistently reported feeling more volitional in their learning (autonomy) and more effective in their language skills (competence) when using ChatGPT. This was particularly evident in areas that are traditionally challenging for EAP students, such as academic writing and pronunciation, aligning with the findings of Xiao and Zhi [13] and Tseng and Lin [15].

However, our study uniquely highlights the complex role of relatedness in AI-assisted EAP learning. While some students experienced a sense of connection with ChatGPT, others felt isolated from human interaction. This variation in relatedness satisfaction is a novel finding in the context of AI-assisted language learning and suggests that the impact of AI tools on social connectedness in language learning is more complex than previously understood [22,26].

### 5.2. Theoretical Significance

These findings have significant implications for SDT in the context of AI-assisted language learning. They support the theory's emphasis on the interdependence of the three basic needs [4,21] but also suggest that this interdependence may manifest differently in AI-mediated environments.

The strong support for autonomy and competence is consistent with the postulation of SDT that environments supporting these needs foster more self-determined forms of motivation [23,24]. However, our findings on relatedness challenge the straightforward application of SDT to AI contexts. The varied experiences of relatedness suggest that AI tools like ChatGPT may satisfy this need for some learners while thwarting it for others, indicating a need for a more comprehensive understanding of relatedness in AI-assisted language learning environments, as suggested in previous research [25–28].

Furthermore, our study extends SDT by exploring how these needs interact in an AI-assisted EAP context. The finding that ChatGPT can simultaneously support auton-

omy and competence while having variable effects on relatedness suggests a need for further theoretical development on how the basic needs are integrated and prioritised in technology-enhanced learning environments, building on the work of Noels et al. [23] and Alamer [24].

The interaction between autonomy, competence, and relatedness in influencing motivation appears to be more complex in AI-assisted learning environments than in traditional contexts. For instance, the strong support for autonomy and competence provided by ChatGPT seems to motivate some students despite the potential lack of relatedness. This suggests that in AI-assisted language learning, the weights of these needs in determining overall motivation might shift, with autonomy and competence potentially compensating for reduced relatedness in some cases. However, for other students, the lack of relatedness appears to undermine the motivational benefits gained from increased autonomy and competence. This highlights the need for a more dynamic model of need satisfaction in AI-assisted learning contexts within the SDT framework.

Finally, our findings also resonate with Dewaele's [32] work on Foreign Language Enjoyment (FLE). The present study suggests that ChatGPT can promote FLE by providing a user-friendly, non-judgmental environment for language practice, particularly for learners with lower levels of proficiency or who are relatively shy. This increased enjoyment may contribute to the satisfaction of learners' basic psychological needs, supporting their feelings of competence and autonomy in their language-learning journey. This connection between ChatGPT use, FLE, and the satisfaction of basic psychological needs provides a new perspective on how AI tools can enhance the affective aspects of language learning, complementing the cognitive benefits often associated with such technologies.

### 5.3. Practical Implications

Our findings have several practical implications for EAP educators and curriculum designers:

1. Integrating ChatGPT to Support Autonomy and Competence

The findings strongly indicate that ChatGPT supports students' autonomy and competence in EAP learning. Thus, educators should:

- incorporate ChatGPT as a supplementary tool for self-directed learning, allowing students to explore language concepts at their own pace;
- design assignments that encourage students to use ChatGPT for brainstorming, drafting, and self-editing, promoting autonomy in the writing process;
- use ChatGPT to provide personalised feedback and examples, supporting students' sense of competence in tackling complex language tasks.

2. Balancing AI and Human Interaction

Given the mixed impact on relatedness, educators should:

- create blended learning environments that combine ChatGPT use with peer-to-peer and student–teacher interactions;
- implement collaborative projects where students share and discuss their ChatGPT-assisted work, fostering a sense of community;
- regularly facilitate in-class discussions about students' experiences with ChatGPT, addressing both benefits and challenges.

3. Developing AI Literacy and Critical Thinking

To address concerns about accuracy and overreliance, educators should:

- integrate lessons on AI literacy, teaching students how to critically evaluate ChatGPT's responses;
- encourage students to verify information from ChatGPT using academic sources, promoting critical thinking skills;
- guide students in developing effective prompts to elicit more accurate and relevant responses from ChatGPT.

4. Scaffolding Relatedness in AI-Assisted Learning

To enhance relatedness:
- Create (online) forums groups where students can share their ChatGPT experiences and support each other.
- Assign peer-review tasks where students use ChatGPT to provide feedback on each other's work, combining AI assistance with peer interaction.
- Implement a mentoring system where more experienced ChatGPT users guide newcomers, fostering a sense of community.

5. Addressing Language Anxiety and Building Confidence

Using ChatGPT's potential to reduce language anxiety:
- Encourage students to use ChatGPT for low-stakes practice before engaging in class discussions or presentations.
- Design activities where students can rehearse conversations with ChatGPT before participating in real-life academic discussions.
- Use ChatGPT to generate personalised confidence-building exercises for students struggling with specific language aspects.

6. Promoting Learner Autonomy beyond ChatGPT

To prevent overreliance:
- Teach strategies for independent language learning that complement ChatGPT use.
- Gradually reduce scaffolding, encouraging students to rely more on their own skills as they progress.
- Implement reflective practices where students regularly assess their reliance on ChatGPT and set goals for independent learning.

7. Customising ChatGPT Use for Individual Needs

Recognising the varied experiences reported:
- Conduct regular assessments to understand individual students' needs and preferences in using ChatGPT.
- Offer flexible options for ChatGPT integration, allowing students to choose how much they incorporate it into their learning.
- Provide additional support or alternative resources for students who find ChatGPT less beneficial or engaging.

### 5.4. Challenges and Limitations

This study identified potential challenges, such as the risk of overreliance on ChatGPT and concerns about the accuracy and relevance of its responses. These challenges highlight the importance of developing students' critical-thinking skills and ability to evaluate the quality of AI-generated content [33,34]. Based on the findings of Yan's [35] study, educators are advised to approach the integration of this technology from a balanced and well-informed perspective, acknowledging both the strengths and the limitations of ChatGPT.

The mixed results regarding the satisfaction of relatedness needs highlight the complex nature of AI-assisted language learning. While some participants experienced a sense of companionship and guidance in their interactions with ChatGPT, others expressed concerns about reduced human interaction and lack of emotional connection. These findings add to our understanding of the potential trade-offs between AI-assisted learning and the satisfaction of psychological needs [28].

From a critical perspective, it is important to acknowledge the potential for ChatGPT to perpetuate or even amplify existing biases and inequalities in language education [10]. As an AI system trained on large datasets of human-generated text, ChatGPT may reflect and reproduce the biases and cultural assumptions embedded in those datasets. Another challenge is the potential for ChatGPT to undermine the development of language skills and competencies that require human interaction and authentic communication.

Furthermore, the integration of ChatGPT into language education raises important ethical questions around issues such as data privacy, intellectual property rights, and academic integrity. Educators and institutions must develop clear policies and guidelines around data protection and ensure that students are aware of their rights and responsibilities when using AI tools like ChatGPT.

## 6. Conclusions and Future Research

This study has examined how ChatGPT influences EAP students' basic psychological needs for autonomy, competence, and relatedness. Our findings suggest that ChatGPT can enhance students' sense of autonomy and competence in language learning, providing flexible, personalised learning experiences and immediate feedback. However, the impact on relatedness was mixed, with some students experiencing a sense of companionship with the AI, while others felt disconnected from human interaction.

These results illustrate the complex relationship between AI-assisted learning tools and students' motivational needs. While ChatGPT shows promise in supporting certain aspects of motivation, it also introduces new challenges, particularly in sustaining social connections and avoiding over-reliance on AI tools.

Our study contributes to research on AI in education by offering a detailed analysis of how AI tools interact with learners' psychological needs in EAP contexts. It also provides insights into the experiences of international students using AI for language learning in an English-speaking country, addressing both their academic and their daily life language needs. This study expands the application of SDT theory to AI-assisted language learning, indicating that the conventional understanding of basic needs' interaction may require reconsideration in these new settings, particularly for learners navigating a foreign academic and cultural environment.

Future research should explore the use of ChatGPT in a wider range of EAP contexts, including undergraduate programmes, and in different geographical and cultural settings. A mixed-methods approach, combining interviews with other data sources, such as learning analytics, classroom observations or linguistic analysis of student–ChatGPT interactions, could provide a more comprehensive understanding of the impact of ChatGPT on student learning and motivation.

Additionally, future research could benefit from an approach that addresses both human–AI and human–human aspects of relatedness in the context of AI-assisted language learning. Longitudinal studies that track the impact of ChatGPT use over time, as well as comparative studies that explore the relative strengths and weaknesses of different AI tools for EAP learning, would be valuable.

Finally, ongoing research is needed to explore the ethical dimensions of AI use in language education to ensure that AI tools such as ChatGPT are used in ways that are equitable, transparent, and consistent with the values of academic integrity and student well-being. This could involve investigating the long-term effects of AI-assisted learning on students' language development and critical-thinking skills.

**Author Contributions:** Conceptualization, J.D. and A.A.; methodology, J.D. and A.A.; data collection, J.D.; formal analysis, J.D. and A.A.; data curation, J.D.; writing—original draft preparation, J.D. and A.A.; writing—review and editing, A.A. and J.D.; supervision, A.A.; funding acquisition, A.A. All authors have read and agreed to the published version of the manuscript.

**Funding:** This research received no external funding.

**Institutional Review Board Statement:** The study was conducted in accordance with the Declaration of Helsinki and approved by the Institutional Review Board (or Ethics Committee) of the University of Otago (Reference: 24-037; Approval Date: 19 April 2024).

**Informed Consent Statement:** Informed consent was obtained from all subjects involved in the study.

**Data Availability Statement:** The raw data supporting the conclusions of this article will be made available by the authors on request.

**Conflicts of Interest:** The authors declare no conflicts of interest.

### Appendix A  Mapping of Interview Questions to SDT Constructs

| Interview Question | SDT Construct |
| --- | --- |
| Can you briefly share your background and experience with English for Academic Purposes (EAP) language learning within higher education in New Zealand? | General |
| How familiar are you with generative AI, specifically ChatGPT, as a language-learning tool? | General |
| What are your initial thoughts or perceptions about using generative AI-ChatGPT as a supporting tool for EAP language learning? | General |
| What is your experience of ChatGPT in improving your English language competence? Please elaborate with examples. | Competence |
| How does the ChatGPT improve your capacity to complete EAP-learning tasks? | Competence |
| Are there any concerns or challenges you foresee in using generative AI like ChatGPT for EAP language learning? | General |
| Do you feel more socially connected or more isolated when you interact with the ChatGPT for EAP language learning? | Relatedness |
| In your opinion, what could be the overall impact of incorporating generative AI like ChatGPT into EAP language learning within the higher education system in New Zealand? | General |
| Can you describe how using ChatGPT has influenced your sense of control and choice in your EAP-learning activities? | Autonomy |
| How has using ChatGPT affected your confidence in your ability to engage with and complete EAP-learning tasks? | Competence |
| Can you share any experiences where using ChatGPT has made you feel more connected to your EAP learning community or has supported your sense of belonging in the EAP classroom? | Relatedness |
| Are there any ways in which using ChatGPT has made you feel less in control of your EAP learning or has limited your choices in how you engage with EAP-learning activities? | Autonomy |
| Can you describe any instances where using ChatGPT has undermined your confidence in your EAP-learning abilities or has made you doubt your competence in engaging with EAP tasks? | Competence |
| Are there any aspects of using ChatGPT that have made you feel less connected to your EAP-learning community or have undermined your sense of belonging in the EAP classroom? | Relatedness |

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
