# Peer review of "The Impact of ChatGPT on English for Academic Purposes (EAP) Students’ Language Learning Experience: A Self-Determination Theory Perspective"

_education, doi:10.3390/educsci14070726_

Round 1
Reviewer 1 Report
Comments and Suggestions for Authors
Dear author(s)
Thank you for this crucial and timely manuscript in which you have tried to reveal the pluses and possible drawbacks in using ChatGPT.
As you have shown, though based on a small group of surveyed students, the pros and cons are almost balanced. Of course, more thorough research are necessary to find the golden mean.
Author Response
Thank you so much for your feedback. The authors appreciate the professional comments from the reviewer's report. We thank you for the comments and for the time you dedicated to reviewing the paper.

Reviewer 2 Report
Comments and Suggestions for Authors
The manuscript entitled “The Impact of ChatGPT on EAP Students' Language Learning Experience: A Self-Determination Theory Perspective” provides insights into learners’ motivational reflections toward using ChatGPT on their EAP learning. In general, I would suggest the author(s) enhance the clarity when explaining and giving examples regarding the three dimensions of SDT. Moreover, more attention may be paid to EAP learning context when quoting interviewees’ responses. Detailed suggestions are provided as follows for the author(s)’ reference.
1.In Section 2.1, the studies cited were mainly from teachers’ perspectives, including teachers’ perceptions towards ChatGTP regarding its benefits and challenges, and how teachers integrate ChatGTP into their pedagogical design. However, the current study focused on students’ perceptions, it would be better to pay more attention to the literature in which learners’ voices were discussed.
2.In Section 2.3, How did the author(s) operationalize the concepts of the three dimensions of SDT? I would suggest the author(s) explain the operational definitions of the main concepts in Section 2.3.
3. Regarding relatedness, the author(s) refer to Ryan & Deci (2017) and define it as “the need to feel connected to others, to have a sense of belonging, and to care for and be cared for by others” in P.3, Line 108-109. Jeon (2022) (cited in the Reference [23]), on the other hand, only highlighted the connection between learners and learning APP, the items in the construct of perceived relatedness in their survey were: I receive support from this app; This app provides me with meaningful information that I can rely on; I feel attached to and comfortable with this app. In the current study, the author(s) indicated that “Relatedness concerns the need to feel connected, belonging, and cared for, both in their interactions with ChatGPT and in their wider social context.” (in P.8, Line 341-342,). However, in the interview questions listed in Appendix A, only peer- or student-teacher interaction was discussed, with neglect of learner-ChatGTP interaction. It would be better if the author(s) clarified this concept and made consistency throughout the manuscript.
4. In Section 3.1, more information about the participants may be needed to help readers understand the transferability of the research (please refer to Shenton, A. K. (2004). Strategies for ensuring trustworthiness in qualitative research projects. Education for information, 22(2), 63-75.). Only the duration of using ChatGTP was considered, what about the frequency? The participants may be in different school years from different institutions, and thus they may have different EAP aims and purposes, and use ChatGTP to conduct different EAP learning activities. All these factors may impact their perceptions. Detailed information about the participants, especially how they integrated ChatGTP into their EAP learning activities may provide a deeper understanding of their motivations and perceptions.
5. In Section 4.1, P15, P16, and P22 simply listed the advantages of AI-tool in their English learning, without an emphasis on EAP.
6. I noted that the manuscript mainly discussed learners’ use of ChatGPT for their English learning beyond the classroom or in their self-directed learning context. It would be better if there was a stronger connection between the research findings and the pedagogical implications provided in Section 5.
Round 2
Reviewer 2 Report
Comments and Suggestions for Authors
Dear Author(s),
Thank you for your efforts to address my comments and suggestions. I would like to suggest an Accept in present form.
Author Response
Thank you so much for your work. I really appreciate that. I am so glad you would like to suggest an Accept in present form.